# A Study on the Behavior of Cadmium in the Soil Solution–Plant System by the Lysimeter Method Using the ^109^Cd Radioactive Tracer

**DOI:** 10.3390/plants12030649

**Published:** 2023-02-01

**Authors:** Vyacheslav Anisimov, Lydia Anisimova, Dmitry Krylenkin, Dmitry Dikarev, Andrey Sanzharov, Yuri N. Korneev, Ilya Kostyukov, Yuri G. Kolyagin

**Affiliations:** 1Russian Institute of Radiology and Agroecology, Kievskoe sh., 109th km, Kaluga Region, 249032 Obninsk, Russia; 2Faculty of Chemistry, Lomonosov Moscow State University, Leninskie Gory, 1, 119991 Moscow, Russia

**Keywords:** Cd, barley, flow-through lysimeter, migration, parameters, forms, mass (volumetric) activity density, specific activity, high-molecular-weight dissolved organic matter, NMR

## Abstract

In soils, cadmium (Cd) and its compounds, originating from industrial activities, differ both in mobility as well as in their ability to permeate the soil solution from naturally occurring cadmium compounds (native Cd). Therefore, the determination of the parameters of cadmium mobility in soils and its accumulation by plants in the soil–soil solution–plant system is very important from both scientific and practical viewpoints. ^109^Cd was used as a radioactive tracer to study the processes of the transition of Cd into the aqueous phase and its uptake by plants over the course of a vegetative lysimeter experiment. Using sequential extraction according to the Tessier–Förstner procedure and modified BCR schemes, certain patterns were determined in the distribution of Cd/^109^Cd among their forms in various compounds in the soil, along with the coefficients of the enrichment of native stable Cd with radioactive ^109^Cd. It was shown that the labile pool of stable Cd compounds (29%) was significantly smaller than that of radioactive ^109^Cd (69%). The key parameters characterizing the migration capacity of Cd in the soil–soil solution–plant system were determined. It was found that the distribution coefficient of native Cd between the soil and the quasi-equilibrium lysimeter solution exceeded the similar value for the ^109^Cd radionuclide by 2.2 times, and the concentration coefficients of Cd and ^109^Cd in the barley roots were 9 times higher than in its vegetative parts. During the experiment, the average removal of Cd (^109^Cd) from the soil by each barley plant was insignificant: 0.002 (0.004)%. Based on the results of ^13^C nuclear magnetic resonance (NMR) spectroscopy of a lyophilized sample of the high-molecular-weight dissolved organic matter (HMWDOM) of the soil solution, its components were determined. It transpired that the isolated lyophilized samples of HMWDOM with different molecular weights had an identical structural and functional composition. The selective sorption parameters of the HMWDOM and humic acid (HA) with respect to Cd^2+^ ions were determined by the isotope dilution method.

## 1. Introduction

Cadmium (A_r_ = 112) is a trace element (the world average concentration in soils varies from 0.2 to 1.1 mg/kg (mean 0.41 mg/kg). For uncontaminated soils, its content varies from 0.01 to 0.3 mg kg^−1^ [1]). According to its geological classification, Cd belongs to the group of chalcophiles. In humus soils, cadmium can act as an organophile, and in some ortsteins as a manganophile [2]. In chemical properties it is close to zinc, but unlike that element, it is characterized by greater mobility [3].

Some sources of cadmium pollution of ecosystems are gas and dust emissions into the atmosphere through various facilities, especially non-ferrous metallurgy (up to 60% of the total metal intake into the soil), fuel combustion and incineration of municipal waste [3], and waste water from mines and industrial enterprises, including small factories. The latter source of cadmium pollution is especially relevant for tropical and subtropical regions of Southeast Asia [4]. In aerosol precipitation, Cd is mainly represented by oxides (up to 71%) [3]. It was found that in solid fine atmospheric particles with a diameter of 2.5 to 10 μm (PM_2.5_–PM_10_), the proportion of Cd mobile forms (including those extracted with weak acetic acid) reaches 50% [5].

The major factor influencing the immobilization of cations of heavy metals (HM, including Cd) in soils is their sorption onto the active soil surface components by nonspecific (electrostatic) and specific (covalent) forces [6]. In addition, Cd could be fixated in soils as a result of precipitation, coagulation, and sorption by clay minerals [3]. These processes result in a wide range of Cd compounds in the soil, which can be classified into groups called operational forms. They could be fractionated by single or sequential extraction, using reagents that differ in the degree of their destructive effects on the soil matrix [3,7,8,9,10]. This grouping of HM compounds in soils, though vulnerable to criticism, is very convenient in terms of practical use, since it allows for separating HM compounds according to their mobility and bioavailability in the soil-soil solution-plant system. 

The works of N.G. Zyrin et al. and other researchers [3,11,12] have shown that most of the Cd in soils (43–84%) is accumulated in mobile fractions, including exchangeable and carbonate-bound forms. Due to such a high Cd content in mobile fractions, high accumulation ratios of this highly toxic metal by plants are observed.

The relative content of various forms of Cd in soils, estimated by Tessier’s sequential extraction procedure (SEP), varies within a wide range: fraction I (exchangeable) 9–55%; fraction II (bound to carbonates) 7–28%; fraction III (bound to iron and manganese oxides) 15–29%; fraction IV (bound to organic matter and sulfides) 1–14%; fraction V (residual) 6–40% [9,11,13,14].

Interesting results on the dynamics of the forms of Cd applied as a water-soluble salt to 0–20 cm topsoil originating from an agricultural land in the Tongguan Mining Area of Shaanxi Province (China) are presented in this study [14]. For 120 days, the authors observed the transformation of Cd operational forms under various water management regimes. It was found that with soil moisture equal to field capacity (FC), the relative content of the exchange form of Cd practically did not change, varying in the range of 39–46% of the total metal amount in the soil; the amount of Cd associated with carbonates decreased from 44 to 9%, the amount bound to Fe-Mn-oxides increased from 7 to 30%, and the amount bound to organic matter and sulfides increased from 0.9 to 17%. The amount of residual Cd practically did not change, varying in the range of 0.8 to 1%.

The BCR technique gives us the following distribution of Cd in soils: fraction 1 (exchangeable and carbonate-bound) 16–42%; fraction 2 (reducible, Fe/Mn oxide-bound) 23–48%; fraction 3 (oxidizable, sulfide/humus-bound) 2–17%; and fraction 4 (residual) 6–57% [11,12,15].

An important indicator characterizing the mobility of pollutants in the soil (their ability to enter the soil solution or groundwater and be absorbed by plant roots) is the size of the pool of mobile compounds of the pollutants (including HM) in the soil (*E*-value) [16,17,18,19]. To determine this indicator, the isotope dilution method is applied, with the use of radioactive or stable isotopes of the elements under study. Since the isotopes added into the soil in ultra-micro concentrations have practically no effect on its composition, the isotope dilution method may be considered optimal for estimating the pool of mobile (*E*-value) HM compounds in soils. The essence of this method as applied to soil studies is to quantify the sum total of potentially mobile (labile) forms of the native isotope of the studied pollutant (for example, *Cd) by measuring the change in the ratio of **Cd/*Cd in an equilibrating solution with a known ratio of the tracer (**Cd) and the investigated isotope *Cd, upon attaining equilibrium in the soil–solution system. The isotope dilution method, though a source of valuable data on the size of the stocks (pools) of mobile and bioavailable HM compounds in the soil, does not provide information on the potential mobility in the soil–soil solution system and the bioavailability of various HM forms. To compensate for this shortcoming, it is necessary to use an integrated approach which will incorporate both the isotope dilution method and sequential extraction procedures for the soil, into which a stable or radioactive HM label has been previously added.

The content of the pool of labile Cd compounds could vary significantly depending on soil properties and the nature of the anthropogenic fallout. Expressed as a percentage of gross content in soils, the *E_Cd_* value ranges from 20 to 70% for the upper (0–20 cm) horizons (including plowed ones) of different soils. Gray et al. [20] give *E_Cd_* values for 13 soil samples taken from various agricultural plots located in New Zealand and belonging to the same soil type (Entisol). Another thing that all the samples had in common was the fact that sewage sludge had been added to the soils they were taken from on a regular basis for a long time. The *E_Cd_* values obtained by the authors varied in the range of 33–55%. Chinese researchers [15] found that in two soil samples taken from a layer of 0–10 cm on relatively slightly Cd-contaminated agricultural lands in China, the *E_Cd_* values varied in the range of 24–30%, and for the soil polluted with waste from enterprises producing ceramic products, the *E_Cd_* value was 74%.

Garforth et al. [18] showed that the *E_Cd_* value for topsoil samples (0–20 cm) of two relatively uncontaminated UK soils varied in the range of 46–47%, while the *E_Cd_* values for two Cd-contaminated soils (the total Cd contents were 47.7 and 64.4 mg kg^−1^) were 34 and 60%, respectively. Interesting data on the dependence of the *E_Cd_* value on the degree and genesis of soil contamination with Cd are presented in the work of French researchers [17]. It was found that in the samples of eight quasi-background rural soils (total Cd vary from 0.1 to 0.89 mg kg^−1^), *E_Cd_* values varied in a wide range of 33–55%. In the moderately contaminated soils (with total Cd contents varying in the range 1.7–6.0 mg kg^−1^), *E_Cd_* varied from 14% (soils from home gardens) to 39–46% (soils from fields irrigated with urban waste waters). In the soil industrially contaminated by the atmospheric fallout of a Pb–Zn smelter, the total Cd content and *E_Cd_* for upper horizon L were equal, respectively, at 21 mg kg^−1^ and 57%, and for underlying horizon B were 19 mg kg^−1^ and 33%. For the soil artificially contaminated by applying increasing amounts of water-soluble CdSO_4_ to obtain total Cd contents in the range 0.54–51mg kg^−1^, the values of *E_Cd_* varied from 54 to 99% after incubation for 4 months, increasing with the growth of Cd concentration. For soils enriched in geogenic Cd (from calcareous parent materials) the *E_Cd_* values were low and equal, on average, to 20%.

Migration of Cd with infiltration waters into the lower horizons of the soil could be observed under conditions of the udic moisture regime [21]. The concentration of Cd in soil solutions lies in the range of 0.2–6 µg dm^−3^ [9]. In natural waters, the proportion of complex Cd compounds with organic matter reaches 90% [22]. According to numerous literature data [23,24,25,26,27,28,29], HM, including Cd, are present in soil solutions mainly in the form of compounds associated with water-soluble organic matter, which can be divided into low- and high-molecular-weight dissolved organic matter. The low-molecular-weight dissolved organic matter (LMWDOM) predominant in soil solutions is the product of the vital activity of soil living organisms and the decomposition of organic substances [24,25]. For example, Knoth de Zarruk [23] showed that low molecular fractions DOM < 3.5 kDa contain up to 50% of the total amount of dissolved organic carbon (DOC). In another work [27], it was noted that fractions of DOM < 5 kDa in solutions are present in the form of “true” solutions, while fractions > 5 kDa are present in the form of colloidal particles.

The accumulation of Cd in plants depends on the concentration of its mobile forms in the soil. In uncontaminated soils, its content in dry plant biomass and grain is in the range of 0.01–0.9 mg kg^−1^ [30,31,32]. It is known that Cd is a toxicant, carcinogen and teratogen [9,32,33].

It is considered that cadmium acts as a natural antagonist of zinc, preventing the latter from plant uptake [9,30]. It has been established that Cd plant uptake, followed by its translocation to the vegetative parts, occurs according to the barrier type [34]. High phytotoxicity of cadmium is explained by its similarity in chemical properties to zinc. Therefore, cadmium can block the action of the zinc-cofactor in many biochemical processes, disrupting the work of the most important Zn-containing enzymes. It has been established [35] that Cd, competing with Zn, Ca and Fe, deactivates cofactors of cell carrier proteins, enzymes of the electron transport chain (thus inhibiting the flow of electrons that can react with reactive oxygen species—ROS), depletes the pool of reduced glutathione due to increased affinities of cadmium to thiol groups [36,37].

Due to the high lability and phytotoxicity of Cd in soils, it is important to be able to assess in advance the potential harm that could be caused to ecosystems by anthropogenic pollution with this dangerous pollutant. An excellent help in this matter is the implementation of semi-mechanistic models of the behavior of HM in the soil–soil solution–plant system the quality of which is determined by the degree of reliability of the key parameters, such as: the proportion of mobile forms of HM in the soil, the distribution coefficient between the solid and liquid phases, the concentration factor (the ratio of HM concentrations in the plant biomass and solution phase), etc.

The aim of this work was to find out the patterns and obtain the parameters characterizing the mobility in the soil and uptake of native cadmium by plants in the soil–soil solution–plant system using the ^109^Cd radioactive tracer in the course of a lysimeter experiment.

## 2. Results

### 2.1. Physico-Chemical Properties of the Soil

The native soil studied for the purposes of this work is characterized by a slightly acidic reaction (pH_KCl_ 5.05 ± 0.01, pH_water_ 6.04 ± 0.01); low hydrolytic acidity (H_g_ = 1.89 ± 0.02 cmol (+) kg^−1^ of soil); sandy loam granulometric composition (fraction < 0.01 mm is 18.5%); very low organic carbon content (1.0 ± 0.01%); low exchangeable potassium (77.7 ± 1.3 mg K_2_O kg^−1^) and a high content of mobile phosphorus (126.9 ± 1.9 mg P_2_O_5_ kg^−1^). The amount of exchangeable cations (Ca^2+^, Mg^2+^, K^+^) was equal to 5.70 ± 0.68, 0.53 ± 0.05 and 0.19 ± 0.02 cmol (+) kg^−1^ of soil, respectively. The mass fraction of the total content of Cd was 0.24 ± 0.02 mg kg^−1^.

### 2.2. Chemical Composition of a Lysimeter Solution

The general chemical composition of quasi-equilibrium lysimeter waters, expressed by Kurlov’s formula [38] at the end of the vegetation experiment, was as follows:(1)$$M_{0.65}NH4_{3.8}Sr_{0.51}Mn_{0.08}Fe_{0.07}Cu_{0.025}Zn_{0.024}\frac{CO_{3}81NO_{3}19}{Ca71\left(K+Na\right)20Mg8},\;pH5.61$$

In this expression, the percent-equivalent fractions (% eq.) of the main anions (whose fraction is more than 10%) are given in descending order in the numerator of the pseudo-ratio, and of the cations in the denominator. On the right-hand side of the formula is the pH value, and on the left-hand side are the mass fraction of the dry residue (g dm^−3^) and, in descending order, the mass fractions of minor cations in the lysimeter solution (mg dm^−3^). In accordance with this hydro-chemical formula, the lysimeter solution can be classified as slightly acidic calcium bicarbonate.

The nitrate content in lysimeter waters is very high (125 mg dm^−3^), which is three times higher than the MPC_NO3_ in water (for reservoirs of fishing importance). This is obviously due to the negative sorption of nitrate ions by the soil and their almost complete transition from native soil to solution. Furthermore, in lysimeter waters, the content of ammonium cations, 3.8 mg dm^−3^, was significantly higher than MPC_NH4_ = 0.5 mg dm^−3^. While the concentrations of cadmium isotopes (0.37 μg dm^−3^) were more than an order of magnitude lower than the MPC_Cd_ = 5 μg dm^−3^, the content of trace elements Cu, Mn, Zn and strontium, on the contrary, was higher than the MPC_Me_, Fe < MPC_Fe_. The content of Cl^−^ anions in the lysimeter solution was 5.1 ± 0.5 mg dm^−3^and that of SO_4_^2−^ was less than 1 mg dm^−3^. The specific electrical conductivity was *EC* = 0.62 ± 0.01 dSm m^−1^. The amount of purified water-soluble organic matter in the lysimeter solution, represented by high-molecular-weight compounds (>3.5 kDa), turned out to be insignificant and equal to 3.3 × 10^−3^% of the dry weight.

### 2.3. Mobility Parameters of Cd (^109^Cd) in the Soil-Lysimeter Solution–Plant System 

The content of ^109^Cd in the corresponding objects of study (soil, lysimeter solution, plants) was measured immediately after the completion of the experiment. Then, the obtained values of volumetric activity density of ^109^Cd were calculated back to the beginning of the vegetation experiment.

The moisture content in soil samples from the lysimeter collected immediately after the end of the vegetation experiment with help of the bore sampler was 27.3 ± 0.5%, which is significantly lower than the maximum water holding capacity (30.9 ± 0.3%) but slightly higher than the field capacity (FC): 25.8 ± 1.0%.

Thus, according to the calculations, the mass activity density of ^109^Cd in soil A_m_(^109^Cd_soil_) on the day of the beginning of the vegetation experiment was 497 ± 11 kBq kg^−1^. The volumetric activity density of ^109^Cd in the quasi-equilibrium lysimeter solution at the beginning of the vegetation experiment was 2840 ± 70 Bq dm^−3^. The raw/dry weight of different parts of the individual 14-day plants (mean ± standard deviation) was 120 ± 30/4.00 ± 0.5 mg for the roots and 180 ± 15/16.0 ± 1.5 mg for the vegetative parts (VP).

The most important parameters characterizing the migration process of ^109^Cd (Cd) in the soil–quasi-equilibrium lysimeter solution–plant system are shown in Table 1. They demonstrate the quantitative ratios of ^109^Cd (Cd) between adjacent media (soil and solution) and the various parts of the plants: concentration factors (*CF*), concentration ratios (*CR*), areas of the active (S_active_) and the total (S_total_) root surfaces, and the cation exchange capacity (CEC) of the root absorbing complex (RAC). They enable us to estimate the translocation relationships between various plant parts, as well as the type of sorption of the studied HM by the plants (see the Section 3). One can also estimate the capacity of the ion exchange sites of the apoplast and evaluate their ability to selectively absorb different cations.

The mass activity densities of ^109^Cd (kBq kg^−1^ of dry weight) in plant material (vegetative parts (VP), roots and RAC), measured at the completion of the experiment and then calculated back to its beginning, are given in Table 1. The total amount of stable native Cd in the soil was 250 ± 40 µg kg^−1^, in the quasi-equilibrium lysimeter solution was 0.32 ± 0.12 µg dm^−3^; its amounts in the VP, roots and RAC are given in Table 1.

To find out the contribution of various forms of Cd to the composition of the liquid phase of soils in more detail using ^109^Cd, sequential extraction procedures (SEP) according to Tessier–Förstner and modified BCR [7,8,39] were applied to the soil samples. A comparative analysis of the results of Tessier–Förstner fractionation shows the following ratios of different forms of ^109^Cd /Cd_stable_ isotopes in the soil in percent: F_I_—25.8 ± 1.0 /12.1 ± 0.9; F_II_—43.2 ± 1.3/16.4 ± 2.0; F_III_—27.0 ± 1.0/21.0 ± 1.0; F_IV_—2.0 ± 0.6 /15.2 ± 3.3; F_V_—1.2 ± 0.6/9.5 ± 3.4; F_VI_—0.8 ± 0.1/8.8 ± 1.6; F_VII_—0.2 ± 0.05/18.6 ± 2.5.

The enrichment factor (*EF*) values of the corresponding forms of natural Cd with the radioactive tracer ^109^Cd in relation to the lysimeter solution at the beginning of the vegetation experiment were 1.08 ± 0.07, 1.33 ± 0.11, 0.65 ± 0.02, 0.07 ± 0.01, 0.06 ± 0.02, 0.04 ± 0.01, 0.01 ± 0.00. The value of the A_sp_(^109^Cd/Cd)_solution_ was equal to 3940 ± 70 Bq µg^−1^ (the *EF* (^109^Cd/Cd)_solution_ value is 1.0). It should be noted that the mean value for the soil as a whole, A_sp_(^109^Cd/Cd)_soil_, equals 1990 ± 40 Bq µg^−1^. Therefore, it was lower than the value of A_sp_(^109^Cd/Cd)_solution_. This indicates a greater lability of the radioactive isotope ^109^Cd applied to the soil compared to the stable native Cd.

A comparative analysis of the results of SEP according to the modified BCR scheme revealed the following ratio of different forms of Cd/^109^Cd isotopes in the soil in percentages: F_1_ (exchangeable and carbonate bound)—29.8 ± 8.0/47.5 ± 1.7; F_2_ (reducible, Fe/Mn oxide-bound)—12.9 ± 1.5/34.0 ± 1.4; F_3_ (oxidizable, sulfide/humus-bound)—12.1 ± 4.2/16.6 ± 3.2. 

The *EF* values of the corresponding forms of natural Cd with the radioactive tracer ^109^Cd in relation to the lysimeter solution at the time corresponding to the beginning of the vegetation experiment were 0.86 ± 0.36, 1.36 ± 0.13, 0.72 ± 0.10. 

Based on the results of Tessier-Förstner SEP obtained and the hypothesis about the increasing specificity of extractants used in the sequential extraction method to extract cadmium cations [10], the pool of labile (potentially mobile) Cd (*E*-value) in a unit of soil mass was calculated as a sum of chemical fractions with the ratio *A_sp_*(^109^Cd/Cd)_fr#_/*A_sp_*(^109^Cd/Cd)_solution_ > 1: *E*_Cd_* = Σ*C*(Cd)_FI–II_ = 71 ± 6 µg kg^−1^ (or 28.4 ± 2.2%). The corresponding mass activity density value for ^109^Cd (*A_m_*(^109^Cd)_ΣFI–II_)) at the beginning of the vegetation experiment was equal to 342.8 ± 7.4 kBq kg^−1^ of soil. This value, however, is significantly lower than the real content of labile Cd compounds in the soil (*E*_Cd_ = 126 ± 3 µg kg^−1^ or 50.5 ± 1.2%), calculated by Equation (5). This discrepancy is accounted for by the difference in the extracting abilities of sequentially used reagents between stages II and III of the extraction procedure. It stands to reason that one should take into account this fact while designing a fractionation scheme, with emphasis on evaluating the mobility of HM in the soil.

The energy of Cd^2+^ cations’ binding by soil (and its constituents) is reflected by the value of the distribution coefficients, *K_d_*. The higher the *K_d_* value, the stronger the affinity of the sorbate with the sorbent. The values of *K_d_*(Cd) and *K_d_*(^109^Cd) in lysimeter solution at the end of the vegetation experiment were 830 ± 320 and 370 ± 100 dm^3^ kg^−1^, correspondingly. In our case, *K_d_*(Cd) > *K_d_*(^109^Cd). This indicates a much stronger binding of stable native Cd with the soil-adsorbing complex (SAC) compared to the radioactive ^109^Cd newly introduced into the soil. 

For balance calculations and assessment of the contribution to the process of cadmium isotope exchange between different forms in the soil, data on the removal (*W*) by vegetative parts and barley roots of both natural and radioactive cadmium isotopes from the soil–lysimeter solution system are of particular interest. They can be obtained based on the data in Table 1. The results of removal of stable Cd and radioisotope ^109^Cd per each 14-day plant were as follows:

*W*_barley_(Cd) = *W*_VP_ (Cd) + *W*_roots_(Cd) + W_RAC_(Cd) = (0.007 ± 0.001) + (0.013 ± 0.004) + (0.002 ± 0.001) = 0.023 ± 0.004 µg (single) shoot^−1^, *W*_barley_(^109^Cd) = *W*_VP_(^109^Cd) + *W*_roots_(^109^Cd) + W_RAC_(^109^Cd) = (26 ± 2) + (63 ± 2) + (12 ± 1) = 101 ± 5 Bq (single) shoot^−1^.

### 2.4. The Qualitative Composition of HMWDOM by ^13^C Magic Angle Spinning (MAS) NMR Spectroscopy

The total amount of dry HMWDOM in H^+^ form obtained after dialysis on MEMBRA-CEL 3.5, 7 and 14 kDa membranes was 123, 121 and 85 mg, respectively, i.e., the concentration of the sum of HMWDOM fractions of more than 3.5 kDa, more than 7 kDa and more than 14 kDa, respectively, was 41, 40 and 28 mg dm^−3^.

The relative proportions of various types of carbon atoms in the HMWDOM extracted from soil solution, obtained by integrating the magic angle spinning (MAS) NMR spectra (Figure 1), are listed in Table 2. The N-alkyl and alkoxy fragments contribute mainly (>64% of the total intensity) to the total intensity of ^13^C NMR spectra of all dialyzed and lyophilized samples of the soil solution. Alkyl and anomeric C fragments make significant contributions (12–15%) to the integrated NMR spectrum of the investigated samples. Aromatic fragments are absent in the obtained samples. The contribution of C carboxyl groups does not exceed 8%.

### 2.5. Cd Selective Sorption Parameters by Ca-HMWDOM and Ca-Humate

The parameters of selective sorption of HMWDOM turned out to be very modest compared to similar parameters of the HA extracted from eutrophic peat. Thus, the value of CEC_Cd_ (HA), determined using the isotope dilution method at pH 6.50 ± 0.15, turned out to be equal to 495 ± 64 mmol (+) (100g)^−1^. At the same time, the value of CEC_Cd_ (HMWDOM), determined under similar conditions, turned out to be equal to 113 ± 15 mmol (+) (100g)^−1^.

The most important parameters characterizing the selective sorption of Cd^2+^ cations by the studied sorbents under the conditions of their extremely low concentration in an equilibrium solution (for example, if only the radionuclide ^109^Cd is introduced into the system as a radioactive tracer, as in the current experiment), and under the conditions of the absolute prevalence of Ca^2+^ ions, are the ^109^Cd and Ca distribution coefficients between the sorbents (HMWDOM and HA) phase and an equilibrium solution phase, as well as the selectivity coefficients of the Ca^2+^/Cd^2+^ ion exchange reaction (Equation (6)). Similar scenarios are the most common in the natural environment due to negligible concentrations of Cd dissolved in soil solutions and natural waters (<1 μg dm^−3^). When studying selective sorption, the amount of ^109^Cd adsorbed with HMWDOM was 19,350 ± 270 Bq g^−1^, with a radionuclide content in the equilibrium solution of 40.0 ± 1.5 Bq cm^−3^. For the HA, the corresponding values were 49,100 ± 400 Bq g^−1^ and 2.8 ± 0.5 Bq cm^−3^. Ca adsorbed with HMWDOM was 113 ± 15 cmol(1/2Ca^2+^) kg^−1^, with a metal content in the equilibrium solution of 2 cmol(1/2Ca^2+^) dm^−3^. For the HA, the corresponding values were 495 ± 94 cmol(1/2Ca^2+^) kg^−1^ and 2 cmol(1/2Ca^2+^) dm^−3^.

Based on the above values, the ^109^Cd and Ca distribution coefficients were calculated according to Equations (7) and (8) as *K_d_*(^109^Cd) = 490 ± 16 cm^3^/g, *K_d_*(Ca) = 57 ± 15 cm^3^ g^−1^ (HMWDOM); *K_d_*(^109^Cd) = 17900 ± 2050 cm^3^ g^−1^, and *K_d_*(Ca) = 250 ± 50 cm^3^ g^−1^ (HA). 

The values of *K_s_*^Ca/Cd^ for the compared sorbents were as follows: K_s_^Ca/Cd^(HMWDOM) = 9.0 ± 2.6, *K_s_*^Ca/Cd^(HA) = 75 ± 17. They differ by more than 8 times and this, of course, has a very significant impact on the role of the HMWDOM under consideration in the water-migration processes of HM translocation in soils.

## 3. Discussion

During the research, it was found that cadmium (a radionuclide and stable isotope) accumulates mainly in the roots (the ratio of A_m_(^109^Cd)_roots_/A_m_(^109^Cd)_VPs_ = 8.8 ± 1.7, [Cd]_roots_/[Cd]_VPs_ = 8.9 ± 0.8). Accordingly, the values of the concentration factors, *CF*(Cd) and *CF*(^109^Cd), in the roots were 9 and 8 times higher than those in the VPs of the barley (10,800 ± 2100 in comparison to 1220 ± 120 dm^3^ kg^−1^ for Cd and 9400 ± 1400 in comparison to 1210 ± 70 dm^3^ kg^−1^ for ^109^Cd). Such a type of plant response to a toxic xenobiotic element is called a barrier type [34]. Also, a pattern indicating the barrier type of cadmium accumulation by barley is given by the ratios of *CR*(Cd)_VP_ to *CR*(Cd)_root_ (7.4): 14.7 ± 2.8 to 2.0 ± 0.4 dm^3^ kg^−1^_,_ and *CR*(^109^Cd)_VP_ to *CR*(^109^Cd)_root_ (8.6): 29.2 ± 4.0 to 3.4 ± 0.2 dm^3^ kg^−1^ (Table 1). 

The important parameters for assessing the contribution of a particular form of cadmium to the soil—solution—plant migration chain are the values of the specific activity of ^109^Cd in the studied objects (lysimeter solution, VPs and roots of barley) in terms of the stable Cd contained in the objects, Bq mg^−1^: *A_sp_*(^109^Cd/Cd)_solution_, *A_sp_*(^109^Cd/Cd)_VP_, *A_sp_*(^109^Cd/Cd)_root_ and *A_sp_*(^109^Cd/Cd)_RAC_. The value of the corresponding parameter at the end of the experiment for the quasi-equilibrium lysimeter solution was 3940 ± 70 Bq µg^−1^; the rest of the values are given in Table 1.

Since the aggressiveness (the degree of destructive effect on soil components) of each successive chemical extractant in SEP schemes increases [3,10], less labile Cd compounds are extracted at each subsequent stage. The degree of their lability can be determined by the value of the enrichment factor of the native Cd with the radioisotope ^109^Cd. With the help of Tessier-Förstner SEP, it was found that the relative content of exchangeable, mobile, bound-to-carbonates and bound-to-easily-reducible-Mn (partially Fe) oxide forms of ^109^Cd in the soil (fractions F_I-III_) exceeds the content of the corresponding forms of stable (native) Cd isotope by 2.1, 2.6 and 1.3 times, respectively. At the same time, the values of the relative content of conservative (fractions F_IV_, F_V_) and firmly fixed forms of ^109^Cd in the soil (fraction F_VI_) were significantly lower than that of stable Cd by 7.6, 8.2 and 11.2 times, respectively. The ratio of the persentages of ^109^Cd and stable Cd in the residual fraction of F_VII_ was about 0.01.

The actual values of A_V_(^109^Cd)_solution_ and the concentration of [Cd]_solution_ in a quasi-equilibrium lysimeter (soil) solution reflect the contribution of various forms of cadmium in the soil studied. Indeed, if the enrichment factor (*EF*) of any form of Cd/^109^Cd in the soil with respect to the soil solution is >1, i.e., *A_sp_*(^109^Cd/Cd)_Fr.#_ > *A_sp_*(^109^Cd/Cd)_solution_, then the process of ideal isotope exchange of the radionuclide between the corresponding chemical fractions (forms) and the solution is shifted towards the latter. Additionally, ^109^Cd^2+^ ions in the chemical fractions will be replaced by ions of native stable Cd^2+^. If, however, *A_sp_*(^109^Cd/Cd)_Fr.#_ < *A_sp_*(^109^Cd/Cd)_solution_, the reverse process takes place. Thus, through the liquid phase, Cd/^109^Cd are exchanged between competing binding sites which form respective (fraction-specific) compounds with cadmium ions in the solid phase of soils. The theoretical aspects of the process of ideal isotope exchange for different isotopes of an HM in the soil–soil solution system that introduce the mechanism of transformation of the forms of this particular HM in the soil are discussed in detail in the works of Anisimov et al. [39,40].

The values of the *EF* of exchangeable, actually mobile and carbonate-bound chemical fractions of Cd (F_I-II_) with the radioactive label ^109^Cd relative to the soil solution (*EF_Fr.#_* = *A_sp_*(^109^Cd/Cd)*_Fr.#_*/*A_sp_*(^109^Cd/Cd)_solution_) were >1, and for conservative and firmly fixed fractions were (F_III-VII_) < 1.

This means that ^109^Cd^2+^ ions will predominantly be desorbed into the lysimeter solution from F_I-II_ fractions, and, conversely, the F_III-VII_ fractions will adsorb them from the solution, which will gradually lead to a decrease in A_sp_(^109^Cd/Cd) in the former group of fractions and its increase in the latter. The more the value of *EF_Fr.#_* differs from 1, the farther from the state of isotope equilibrium in the quasi-equilibrium soil–soil solution system will be the radioactive and naturally stable isotopes of cadmium, either bound in the forms of corresponding chemical fractions in the soil or present in free form in the lysimeter (soil) solution.

The equilibration period of native Cd compounds and the added ^109^Cd radiotracer in the moist soil before the beginning of the vegetation experiment was half a year. This time period is more than enough for establishing an ^109^Cd/Cd isotope equilibrium for labile compounds of native Cd in the soil. This, however, is not enough time for the less labile compounds of native Cd to come to isotope equilibrium with the introduced radiotracer. This issue suggests the necessity of combining the isotope dilution method with chemical fractionation to assess the dynamics and direction of transformation of the forms of the industrially introduced Cd (and any other pollutant) in the soil under conditions of relatively high humidity and a degree of aeration sufficient to prevent the development of negative gleying processes. Such conditions are observed, for example, in soils at the sites of municipal solid waste landfills. Flow-through lysimeters, similar to the one shown in Figure 2, are an effective tool for attaining the goals of the experiments.

The removal of Cd (^109^Cd) by barley throughout the experiment period was insignificant. It was 0.002/0.004% of their total amount in the soil, respectively. Consequently, the removal of metal by plants cannot have a noticeable effect on the ratio of the forms of cadmium in the soil.

The results of ^13^C MAS NMR spectroscopy of HMWDOM of soil solutions showed that the signals in the region of 48–93 ppm make the greatest contribution to the total intensity of the spectra of all samples. A broad signal in the area of 48–60 ppm was assigned to the methoxyl groups from guaiacyl and syringyl fragments of lignin, as well as to Cα in polypeptides [41]. This signal was also attributed to N-alkyl in amino acids [42,43]. The signal at 71.5 ppm was attributed to carbon atoms in the pyranoside structures of cellulose and hemicellulose [41,44]. Significant contributions (12–15%) to the total intensity of spectra are made by signals in the regions of 0–48 and 93–113 ppm related to methyl, methylene groups of lipids, polyesters, waxes and anomeric carbon atoms of sugars (93–113 ppm), respectively [41,42,43,44]. Signals in the area of 160–190 ppm refer to carboxyl carbon atoms of carboxyl, ester and amide fragments. Their total contribution does not exceed 8%. It should be noted that all three samples have the same composition at a qualitative level and differ only in molecular weight. Thus, with an increase in the pore diameter of the membrane, the molecular weight of the isolated lyophilized HMWDOM samples increases. From the ratios of the intensity of the NMR spectra, the following ratio of the molecular weights of the samples was obtained: m(3.5):m(7):m(14) = 1:1.5:1.7. Additionally, based on the relative integral intensities, it can be argued that the quantitative contribution of all fragments to the content of the samples is approximately the same.

The aims of this study did not include the study of the role of low-molecular-weight organic compounds (LMWDOM) with a particle size < 3.5 kDa in the processes of ion binding and the subsequent water migration of Cd^2+^. In the present study, efforts were focused on elucidating the role of high-molecular-weight DOM (MW > 3.5 kDa) in Cd binding and migration processes in soil solutions. These substances belonged to a pre-colloidal fraction of the soil solution. In the process of concentration, purification by dialysis in the presence of cationite in H^+^-form and lyophilization, HMWDOM coagulated to form colloidal particles and allowed us effectively to deal with the colloidal HMWDOM in H^+^-form.

Subsequently, during the determination of the CEC and selectivity coefficient (*K_s_*^Ca/Cd^), HMWDOM was converted to Cd^2+^ (Ca^2+^) form by long-term dialysis with 0.01 M [Cd^2+^] (or [Ca^2+^]) equilibrating solution (pH 6.50 ± 0.15). In the end, loaded with cadmium (calcium) ions compounds HMWDOM at the same time represented homoionic colloidal and finely dispersed particles in the equilibrium solution, which made it possible to use a robust and reliable isotope dilution method to evaluate their sorption characteristics. Similar procedures were performed for the humic acid samples obtained from eutrophic peat taken as a comparison sample.

The value of HMWDOM CEC was 4.4 times lower, and the selectivity coefficient (Equation (6)) was 8.3 times lower than that of HA. However, even such moderate parameters of Cd^2+^ ions’ selective sorption by HMWDOM at vanishingly low concentrations in the equilibrium system indicate the important role of these components of the soil solution in the processes of cadmium ion transport in the aqueous medium. The dissociation constant of the hydroxocomplex Cd(OH)^+^ of the first stage *K_D,_*_1_, determined in accordance with Equation (9), is 6.76 × 10^−5^. The total dissociation constant of the hydroxocomplex Cd(OH)_2_, *K_D_*_1,2_, equals 4.68 × 10^−9^ (Equation (10)). Substituting the above-stated data into the material balance Equation (11), we obtain that for pH= 6.5, *α*(Cd^2+^) = 2.14 × 10^−2^. To calculate the dissociation constants of counter-ions, for example, Cd^2+^, from adsorbing complexes with functional groups of HMWDOM and HA sorbents (*K_Cd-HMWDOM_*, *K_Cd-HA_*), we have used the Equations (12)–(15). It is necessary, however, to take into account the activity coefficients of Cd^2+^ cations present in the sorbent adsorbing complex in an insignificant amount, and the activity coefficients of the prevailing Ca^2+^ cations. 

Since the equilibrium system contains only ^109^Cd, introduced as a radioactive tracer, whose specific activity (A_sp_^109^Cd), according to the IAEA electronic handbook [45] is 9.602 × 10^13^ Bq g^−1^, then, using the data on the volumetric activity density of ^109^Cd in equilibrium solutions of 0.01 M Ca(NO_3_)_2_ and the mass activity density in the phase of HMWDOM and HA sorbents (see Section 2), vanishingly low molar concentrations of ^109^Cd atoms in the sorbent phase and in equilibrium solutions were calculated: [^109^Cd]_HMWDOM_ = (1.85 ± 0.25) × 10^−12^ mmol g^−1^, [^109^Cd]_solution(HMWDOM)_ = (3.92 ± 0.28) × 10^−15^ mmol cm^−3^; [^109^Cd]_HA_ = (4.69 ± 0.39) × 10^−12^ mmol g^−1^, [^109^Cd]_solution(HA)_ = (2.67 ± 0.46) × 10^−16^ mmol cm^−3^. The activity coefficients of ^109^Cd^2+^ and Ca^2+^ ions in 0.01 M Ca(NO_3_)_2_ solution (I = 0.03 M) are the same and equal to 0.45. Meanwhile, the activity coefficients of cadmium ions (^109^Cd^2+^) in the sorbent adsorbing complex, according to [46], with an equivalent proportion of competing Ca^2+^ (*E*_Ca_) cations approaching 1, also tends to 1.

Using the values of [^109^Cd] and [^109^Cd^2+^] in equilibrium solutions obtained at pH 6.50 ± 0.15, as well as the values of the activity coefficients of Cd^2+^ ions, the following parameters were calculated:The dissociation constant of Cd-HMWDOM complex, determined in accordance with Equation (14), is *K*_Cd-HMWDOM_ = 7.7 × 10^−22^;The dissociation constant of Cd-HA complex (Equation (15)) is *K*_Cd-HA_ = 1.4 × 10^−24^.

Thus, HMWDOM molecules present in the soil solution are able to maintain a low concentration of Cd^2+^ cations that prevents their possible coprecipitation with Fe(III), Mn(IV), and Al hydroxides with an increase in the pH values of natural waters and soil solutions. Consequently, we can confirm the dual function of the considered groups of organic compounds in the soil. The presence of HMWDOM increases the mobility of cadmium in soil solutions; however, due to the presence of HA in abundance in the soil, and due to the insolubility of the latter in aqueous solutions, on the contrary, an effective geochemical humus barrier that prevents the aqueous migration of cadmium is formed.

## 4. Materials and Methods

### 4.1. Design of the Lysimeter Vegetation Experiment 

The model experiment with barley culture was carried out in a closed cycle that included the recovery of gravity runoff from sandy loam sod-podzolic soil with drainage, which passed through vegetative vessels with an aqueous barley culture and then re-entered the soil surface (Figure 2a,b).

The model system (vegetation stand) featured a flow-through lysimeter installation that ensured a gravitational flow of soil moisture into flow-through 3 vegetation vessels with 6 cartridges each. Every cartridge had nylon-mesh bottoms and were filled with coarse sand. Three sprouted seeds were planted in each cartridge. Besides the cartridges, the model system contained a peristaltic pump, a silicone tubing, taps and 3-channel glass adapters, a buffer tank, and a lighting fixture. The flow-through lysimeter was covered with a non-hermetic plexiglass lid with a built-in water-sprinkling device (Figure 2a). The detailed designs of various modifications of the vegetation stand are given in [39,40].

As objects of research, barley (*Hordeum vulgare* L.) of the Zazersky 85 variety was used, as well as sod-podzolic sandy loam soil (Albic Retisol (Loamic, Ochric)), collected near Peredol village, in the Zhukovsky district of the Kaluga region.

The physical and chemical properties of the studied soil have been presented earlier in [39,47]. The content of nitrate ions and ammonium cations in lysimeter waters was determined potentiometrically using ion-selective electrodes.

^109^Cd (T_1/2_ = 461.9 days) was applied to the soil in the form of its work solution (without isotopic carrier) in the amount of 650 kBq kg^−1^, which was prepared from a stock solution of ^109^Cd(II) in 0.5 M HCl (“CYCLOTRON Co., Ltd.”, passport No. C-740-19), with an activity at the time of certification equal to 40.0 MBk ^109^Cd. The soil suspension was thoroughly mixed, and nutrients (N, K) were applied in the form of aqueous solutions of salts (NH_4_NO_3_, and KNO_3_) in the doses recommended during such experiments (N_100_, K_100_). Phosphorus was not applied because its content in the soil was sufficient [40,47]. Next, the soil was dried to an air-dry state, ground and sieved through a 2 mm sieve, then placed in a lysimeter installation, with alternating vertical layers of prepared soil and washed quartz fine–grained (0.4–0.8 mm) sand which acted as a drainage (Figure 2b).

After assembly, the lysimeter installation was covered with a light-tight film, 2.0 dm^3^ of deionized water was poured onto the soil surface in a lysimeter and the soil–solution system was balanced for 5.5 months, periodically returning the water flowing from the lysimeter back to the soil surface and replenishing its losses from evaporation.

Two weeks before the start of the vegetation experiment, a vegetation stand was assembled (Figure 2a), including a lysimeter, vegetation vessels, buffer tank, peristaltic pump, filter and connecting hoses (the scheme of the vegetation stand is given in [39,40]. The amount of water in the system was brought to a working volume of 6.0 dm^3^, and a peristaltic pump was started and left in operation for 2 weeks to achieve equilibrium in the system. The rate of moisture flow through the peristaltic pump was set at 40 cm^3^ min^−1^, thus providing 10 cycles of recovery of the lysimeter solution per day. The duration of the vegetation experiment was 14 days. In vegetative vessels, the mixing of solutions was carried out by bubbling with air. Evaporating moisture (≈3%/day) was compensated daily with deionized water.

### 4.2. Determination of Cd (^109^Cd) Mobility Parameters

Immediately after the end of the vegetation experiment, the forms of Cd (^109^Cd) associated with various organo-mineral components of soils were determined by the sequential chemical fractionation procedure according to Tessier in the Förstner modification [7,8,40], in soil samples taken directly from the lysimeter in 3 replicates. The following forms were identified: fraction I (F_I_)—exchangeable (extracted by 1M CH_3_COONH_4_, pH 7.0 (AAB-7.0); fraction II (F_II_), which was actually mobile and carbonate bound (extracted with 1M CH_3_COONa (pH 5.0); fraction III (F_III_), which was associated with easily reducible Mn (partially Fe) oxides (extracted with 0.1 M NH_2_OH-HCl in 0.01 M HNO_3_); fraction IV (F_IV_), which was associated with moderately reducible Fe oxides (extracted with 0.1 M H_2_C_2_O_4_+(NH_4_)_2_C_2_O_4_ buffer solution at pH 3.2–3.3); fraction V (F_V_), which was bound to oxidizable organic matter and sulfides (extracted with 1M CH_3_COONH_4_ + 6% HNO_3_ after pretreatment twice with a solution of 0.02 M HNO_3_ + 30% H_2_O_2_ (v:v=3:5), pH 2; fraction VI (F_VI_), which was bound to the crystal matrix of primary and secondary soil minerals (extracted with hot reagent water after pretreatment with HNO_3_ (conc.) at t = 95 °C to near dryness); and fraction VII, which was residual (bound to the crystal matrix of primary and secondary soil minerals and not leachable by HNO_3_(conc.) when heated). All reagents used in the study were analytical grade.

Various chemical fractions of Cd (^109^Cd) were also isolated using SEP of BCR. This SEP was described in detail in [8,39].

During the vegetation experiment, a number of migration factors/coefficients for Cd and ^109^Cd were determined:distribution coefficient (*K_d_*), dm^3^ kg^−1^: *K_d_*(Cd) = [Cd]_soil_/[Cd]_solution_, and *K_d_*(^109^Cd) = A_m_(^109^Cd)_soil_/A_v_(^109^Cd)_solution_;concentration factor (*CF*) in vegetative parts (VPs), roots and RAC, dm^3^ kg^−1^: *CF*(Cd) = [Cd]_VP_/_root/RAC_/[Cd]_solution_ and *CF*(^109^Cd) = A_m_(^109^Cd)_VP/root/RAC_/A_v_(^109^Cd)_solution_;concentration ratio (*CR*) in VPs and roots: *CR*(Cd) = [Cd]_VP/root/RAC_/[Cd]_soil_ and *CR*(^109^Cd) = A_m_(^109^Cd)_VP/root/RAC_/A_v_(^109^Cd)_soil_.

Here, A_m_(^109^Cd) _VP, soil_ is the mass activity density of ^109^Cd in VPs and soil; A_V_(^109^Cd) is the volumetric activity density of ^109^Cd in the solution samples; [Cd]_VP_, [Cd]_soil_, [Cd]_solution_ are the Cd concentration in VPs, soil and solution samples; A_m_(^109^Cd)_root/RAC_ and [Cd]_root/RAC_ are the ^109^Cd mass activity density and Cd concentration in RAC and in roots after the removal of the metal cations adsorbed by the RAC (in the apoplast). 

The sizes of the total (S_total_) and active (S_active_ ) root surface areas per unit of root raw weight, as well as the percentage of the active root surface, were estimated using the calorimetric method of D. Sabinin and I. Kolosov [48]. In essence, this method makes use of the basic dye methylene blue (MB), whose positively charged ions in a solution harmless to plants change the blue color intensity of the solution while having been absorbed by plant roots. 

A total of 1 mg of MB covers 1.05 m^2^ of the absorbent surface. It permeates the cells of the epidermis in 90 seconds. After two 90 sec consecutive submergences of plant roots in an MB solution, all of the root surface area (active and inactive) will be covered in the dye. During a third 90 sec dip in the solution, only the active root surface will still absorb the dye. The total root surface area is calculated from the changes in the MB concentrations in the first two baths; the change in its concentration in the third bath with respect to the first two gives us the size of the active root area. The MB concentration is measured spectrophotometrically (λ = 650 nm).

The cation exchange capacity (CEC) of the RAC of the 14 day barley seedlings was estimated by the Brown and Noggle method [49] after two 20 sec consecutive extractions of the cations absorbed by the apoplast (Ca^2+^, Mg^2+^, K^+^, Na^+^ and NH_4_^+^) with a 0.1 M solution of HCl, followed by a washing of the roots with deionized H_2_O.

The isotope dilution method is the most suitable technique for determining the amount of HM in the soil in a potentially mobile (labile) form. It is based on the law of ideal isotope exchange. The isotope exchange reaction is described by the following chemical equation:Cd_soil_ + ^109^Cd^2+^_solution_ ↔ Cd^2+^_solution_ + ^109^Cd_soil_.(2)

The equilibrium constant of the reaction (Equation (2)) is:(3)Keqv=[Cd]solution×Am(C109d)soil[Cd]soil×AV(C109d)solution

Since at equilibrium of the ideal isotope exchange Δ*G*^0^ = 0, as well as Δ*H*^0^ = 0 and Δ*S*^0^ = 0, then the equilibrium constant *K_eqv_* = 1 [39,40,50].

Hence:(4)Am(C109d)soil[Cd]soil=AV(C109d)solution [Cd]solution 

Thus, the amount of labile Cd in the soil (*E*-value) is:(5)ECd=[Cd]soil=Am(C109d)soil×[Cd]solution AV(C109d)solution 
where A_m_(^109^Cd)_soil_ is mass activity density of ^109^Cd and [Cd]_soil_ is the concentration of the native Cd in the soil; whereas A_v_(^109^Cd)_solution_ and [Cd]_solution_ are volumetric activity density of ^109^Cd and concentration of the native Cd, respectively, in the equilibrium lysimeter solution. 

The total amount of Cd in the soil was estimated after ashing at 500 °C and subsequent decomposition of the samples using HCl(conc.) + HNO_3_(conc.) + HF(conc.). The concentration of Cd (^109^Cd) in the roots and VPs of barley plants was determined after ashing at 500 °C [39,40].

Elemental analysis was carried out by atomic absorption methods with electrothermal atomization (ETA-AAS) of solution samples using a QUANTUM-Z spectrometer for Cd and optical emission of inductively coupled plasma using the Liberty II spectrometer (Varian) for the remaining elements (Ca, Mg, K, etc.). The mass (volumetric) activity densities of ^109^Cd in the samples were determined on the scintillation gamma-spectrometric complex Atom Spectra 2 (NPP KB RADAR Ltd) with a NaI(Tl) 40 × 40 mm crystal and Becquerel Monitor software (version 1.0). Energy calibration was performed using 4 spectral lines: ^241^Am—59.54 keV, ^109^Cd—88.03 keV, ^137^Cs—661.66 keV, ^60^Co—1332.49 keV.

In addition to estimating the mass activity density of ^109^Cd, the ratio of the mass activity density of ^109^Cd to the concentration of stable Cd in the corresponding objects of study (lysimeter waters, parts of plants and certain chemical fractions of cadmium in the soil) were calculated. We called this parameter the “specific activity of ^109^Cd/Cd, A_sp_(^109^Cd/Cd)”, (Table 3). The values of the so-called enrichment factor (*EF*) of stable Cd with the radioactive tracer ^109^Cd were also calculated for individual components of the system—the soil, chemical fractions and plants—as the ratio of A_sp_(^109^Cd/Cd)_Fr.#/VP/root/RAC)_ to A_sp_(^109^Cd/Cd)_solution_. For example, *EF*_fr_._#_ = A_sp_(^109^Cd/Cd)_Fr#_/A_sp_(^109^Cd/Cd)_solution_. The specific activity of ^109^Cd/Cd in the studied samples was calculated back to the date of the lysimeter vegetation experiment beginning.

### 4.3. Determination of the Qualitative Composition of HMWDOM

The components of high-molecular-weight dissolved organic matter of a native quasi-equilibrium soil solution were analyzed after purification and lyophilization using ^13^C NMR-spectroscopy. To obtain low-ash samples of the HMWDOM, a large amount of quasi-equilibrium soil solution (more than 10 dm^3^) was extracted from the sod-podzolic sandy loam soil by continuously circulating for a fortnight a large volume (12 dm^3^) of the deionized water through a flow-through lysimeter, similar to the one used in vegetation experiment.

The resulting solution was divided into 3 parts, each of which, due to their large volume (>3 dm^3^), were subjected to sequential purification to remove silty and colloidal particles (using dense blue ribbon paper filters and nylon membrane filters of 3 μm and 1 μm) before final filtration through a 0.45 μm membrane filter. Between successive filtration stages, soil solutions were stored frozen at t = −18 °C.

After that, each of the 3 portions of the resulting filtrate (V = 3.0 dm^3^) were successively:evaporated on a vacuum rotary evaporator to V≈50 cm^3^ (P = 0.1 ÷ 0.2 atm, t = 50 ÷ 55 °C);subjected to a dialysis procedure in order to remove easily soluble salts and low molecular weight DOM, using MEMBRA-CEL dialysis bags made of durable regenerated cellulose with pore size 3.5, 7 or 14 kDa (Figure 3a). As a dialysis solution, a mixture of deionized water and KU-2-8 cationite (H^+^-form) was used in a 25:1 ratio (*v*/*v*) at constant stirring using a magnetic stirrer. The dialysis solution was replaced with a fresh one on a daily basis. The dialysis procedure was carried out for 7 days. The control was carried out by the specific electrical conductivity (*EC*) of the dialysis solution, the values of which decreased from 350 to <10 μS cm^−1^, corresponding to the plateauing of the specific electrical conductivity readings (Figure 3b);lyophilized by AK40 ProfLab (Figure 3c).

As a result, more than 70 mg of lyophilized HMWDOM (Figure 3d) of HMWDOM in H^+^-form (HMWDOM-H) was obtained from each portion of the soil solution filtrate with a volume of 3 dm^3^, the structural and functional composition of which was analyzed by ^13^C NMR spectroscopy at the Faculty of Chemistry of Lomonosov Moscow State University.

Solid-state ^13^C MAS NMR spectra were recorded on a BRUKER AVANCE-II NMR 400 WB spectrometer operating at the frequency of 400.13 (^1^H) and 100.13 MHz (^13^C) using a 4 mm H/X/Y MAS WVT probe (spinning rate is 10 kHz). A 4-pulse TOSS with RAMP cross-polarization pulse sequence was used (T/tr = 2.2412), with a contact time of 2 ms, a recycle delay of 1.5 s and an 8-step phase cycle. For high-power proton decoupling, the SW-TPPM sequence (τ = 8 μs, φ = 15°) was used. A line broadening of 100 Hz was applied to all samples after Fourier transformation.

Regions corresponding to the following types of carbon were identified in the ^13^C MAS NMR spectra of HMWDOM: alkyl groups (0–48 ppm); N-alkyl groups, alkoxy groups (48–93 ppm); anomeric C in O-C-O fragments (93–113 ppm); aromatic C (113–160 ppm) and carboxyl groups (160–190 ppm) [41,42,43,44].

### 4.4. Determination of the Selective Sorption Parameters of Cd by HMWDOM-Ca and Humate-Ca

The potential ability of the HMWDOM to bind Cd^2+^ cations present in the soil solution is determined by the following parameters: cation exchange capacity (CEC), and selectivity coefficient of Ca^2+^/Cd^2+^ exchange, *K_s_*^Ca/Cd^. The CEC value of the lyophilized HMWDOM (3.5 kDa) was estimated by an isotope dilution method [15,18,19,39,51]. To do this, 30 mg of HMWDOM-H^+^ was placed inside a dialysis tube with a “window” covered with a MEMBRA-CEL membrane (3.5 kDa), and 10 cm^3^ of a 0.01 M Cd(NO_3_)_2_ equilibrating solution was poured in (Figure 4a,b). The pH of the solution was 6.50 ± 0.20 (if required, it was adjusted to the desired value using micro quantities of CdCO_3_ and HNO_3_ when prepared). From the outside, 50 cm^3^ of the same equilibrating solution was poured into the outer container, after which dialysis tubes with HMWDOM suspension were submerged in it with the “window” down. The surface of the liquid in both vessels was at the same level (Figure 4b). The assembly was placed on a rotary shaker and shaken for 3 days, replacing the equilibrating solution in the outer container with a fresh one 2 times every 24 hours. This procedure was repeated 3 times (as a result, the pH of the quasi-equilibrium solution gradually increased and, by the end of the procedure of sorbent saturation with Cd^2+^ ions, it was practically matching the initial pH of the equilibrating solution).

After removal of the external solution for the 6^th^ time, it was replaced with 40 mL of an equilibrating solution (0.01 M Cd(NO_3_)_2_, pH 6.50 ± 0.15) with the ^109^Cd radioactive tracer, and the installation was shaken on a rotary shaker for 3 days [18] or, to be exact, until the isotope equilibrium between the external equilibrating solution and the suspension inside the dialysis tube was reliably established. It is noteworthy that, according to the works of [15,19,20], a single day is enough for attaining the isotope equilibrium in a two-component system (s/l). In the state of isotope equilibrium, A_sp_(^109^Cd/Cd) is the same in all parts of the equilibrium system (in solid and liquid phases). Based on the equality of A_sp_(^109^Cd/Cd)_HMWDOM_ = A_sp_(^109^Cd/Cd)_inner solution_ = A_sp_(^109^Cd/Cd)_outer solution_, one could measure the volumetric activities of ^109^Cd in the equilibrium suspension from the inner dialysis tube and in the equilibrium solution from the outer containers, respectively, and, by the difference in volumetric activities, the amount of Cd adsorbed by HMWDOM can be calculated. Dividing this value by the mass of the HMWDOM taken for analyses, we obtained the value of its CEC as to Cd.

A similar method of isotope dilution is also used to determine another important parameter, the selectivity coefficient (*K_s_*^Ca/Cd^) of Ca^2+^/Cd^2+^ exchange, which characterizes the selective sorption of the studied Cd^2+^ ions relative to the Ca^2+^ cations prevailing in natural absorbing complexes. For this purpose, 30 mg of dry HMWDOM-H was placed inside the dialysis tube, then 10 cm^3^ of a 0.01 M Ca(NO_3_)_2_ equilibrating solution (pH 6.50 ± 0.15) was added. Outside the tube, 50 cm^3^ of the same equilibrating solution was poured into the outer container and the system was brought into equilibrium, as described above. Then, after removing the external solution for the 6^th^ time, it was replaced with 40 mL 0.01 M Ca(NO_3_)_2_ with the added radioactive tracer ^109^Cd (pH 6.50 ± 0.15), and the assembly was shaken on a rotary shaker for 3 days. The amount of radionuclide sorbed by HMWDOM was determined by the difference in the volumetric activity densities of the ^109^Cd suspension inside the dialysis tube and the solution in the external container, and then the K_d_(^109^Cd) value was calculated. The selectivity sorption coefficient of Ca^2+^/Cd^2+^ ion exchange, at a vanishingly low concentration of Cd^2+^ in an equilibrium solution, was calculated by the following equation:(6)$$K_{s}^{Ca/Cd}=\frac{K_{d}\left(Cd\right)}{K_{d}\left(Ca\right)}$$
where
(7)Kd(C109d)=Am(C109d)soilAV(C109d)soil solution
(8)$$K_{d}\left(Ca\right)=\frac{CEC_{Cd}}{\left[Ca^{2+}\right]}$$

*CEC*_Cd_ is the cation exchange capacity of the sorbent in relation to Cd^2+^, determined using an equilibrating solution of 0.01 M Cd(NO_3_)_2_ at pH 6.55 ± 0.15 and ^109^Cd as a tracer (*CEC*_Ca_ ≈ *CEC*_Cd_). 

Because we were dealing with a substance (a set of substances) of unknown composition, we decided to use purified humic acid (HA) in H^+^-form obtained from eutrophic peat (correctly: Fibric Histosols Eutric soil [52]) using the classical method with 0.1 M NaOH extraction [53], which has very high sorption characteristics with respect to HM and radionuclides and is used by RIRAE specialists as a component of a biologically active organo-mineral complex (Patent for invention No. 2709737). All the methodological procedures with HA were similar to those described above for HMWDOM. The specific activity of ^109^Cd in the analytes was calculated to the very beginning of the sorption experiments.

Since most of the cadmium ions in soil solutions are hydrolyzed and present in the form of hydroxo-complexes:Cd(OH)^+^ ↔ Cd^2+^ + OH^−^,(9)
Cd(OH)_2_ ↔ Cd^2+^ + 2OH^−^.(10)

The dissociation constant of the hydroxocomplex Cd(OH)^+^ of the first stage *K_D,_*_1_, calculated in accordance with Equation (9). The total dissociation constant of the hydroxocomplex Cd(OH)_2_, *K_D_*_1,2_*,* was calculated according to Equation (10). Then, based on the material balance Equation (11), it is possible to calculate the proportion of free Cd^2+^ ions in the solution, α(Cd^2+^) [54]:(11)$$\alpha \left(Cd^{2+}\right)=\frac{\left[Cd^{2+}\right]}{\left[Cd\right]}=\frac{1}{1+\beta_{1}\left[OH^{-}\right]+\beta_{1,2}{\left[OH^{-}\right]}^{2}}\;,$$
where *β*_1_ = 1/*K_D_*_1_ and *β*_1,2_ = 1/*K_D_*_1,2_.

If we consider the adsorption complexes of the sorbents studied with adsorbed counter-ions (Cd^2+^, Ca^2+^, etc.) as solid solutions, then we can write chemical dissociation equations to determine the dissociation constants of adsorption complexes of counter-ions, for example, Cd^2+^, with functional groups of HMWDOM and HA sorbents (*K_Cd-HMWDOM_*, *K_Cd-HA_*):Cd-HMWDOM ↔ Cd^2+^ + HMWDOM^2−^,(12)
Cd-HA ↔ Cd^2+^ + HA^2−^,(13)
and, respectively,
(14)$$K_{Cd-HMWDOM}=\frac{\left[Cd^{2+}\right]\left[HMWDOM^{2-}\right]}{\left[Cd-HMWDOM\right]},$$
(15)$$K_{Cd-HA}=\frac{\left[Cd^{2+}\right]\left[HA^{2-}\right]}{\left[Cd-HA\right]}.$$

### 4.5. Statistical Analysis

All experimental data were subjected to statistical processing using standard methods using Microsoft Excel software, and the theoretical foundations set out in [55]. The arithmetic averages and standard deviations of the determined parameters were established. The accepted significance level is *p* < 0.05.

## 5. Conclusions

Over the course of the experiment, the chemical compositions of sod-podzolic sandy loam soil, quasi-equilibrium lysimeter solutions obtained from it, and barley test plants grown on these solutions by the hydroponics method of water culture, were studied in detail. In all of the aforementioned components of the soil–lysimeter solution–plant system, the content of stable and radioactive isotopes of cadmium was also estimated.

The comparative labilities of various operational forms of natural Cd and the radioactive tracer ^109^Cd (according to Tessier-Förstner SEP) were determined. The enrichment of exchangeable, actually mobile and carbonate-bound forms of native Cd by radioactive ^109^Cd is 1.08–1.33 times higher than that in the soil solution. At the same time, the enrichment of conservative and firmly fixed forms of Cd in the soil (Fractions III-VII) by the ^109^Cd radionuclide, as compared to the lysimeter solution, was 1.5–150 times lower. The pool of the native labile Cd compounds in the soil was 51%, which attests to the high mobilization potential of the native Cd compounds.

During the lysimeter experiment, important parameters characterizing the migration mobility and bioavailability of Cd in the soil–equilibrium solution–plant system were also determined. These parameters, the distribution coefficients (*K_d_*) of Cd and ^109^Cd between the solid and liquid phases, reflecting the binding strength of Cd cations by soil, and the metal concentration factors *(CF*) in the barley roots and vegetative parts, as compared to the equilibrium lysimeter (soil) solution, are necessary, particularly when developing and carrying out the parametrization of semi-mechanical models of the pollutant root uptake.

The isolated lyophilized samples of HMWDOM soil solution with different molecular weights (>3.5 kDa, >7 kDa and >14 kDa) have the same qualitative composition and differ only in molecular weight.

HMWDOM, despite their low concentration in a quasi-equilibrium soil solution extracted from sod-podzolic soil, play an important role in the processes of cadmium ion transport in the liquid phase, as they also prevent the possible coprecipitation of cadmium ions with Fe(III), Mn(IV), and Al hydroxides with an increase in the pH values of natural waters and soil solutions. However, unlike HMWDOM, humic acids, due to their insolubility in aqueous solutions and high adsorption capacity with respect to Cd^2+^, on the contrary contribute to its immobilization in the soil, thereby forming an effective geochemical humus barrier that prevents the aqueous migration of cadmium.

## Figures and Tables

**Figure 1 plants-12-00649-f001:**
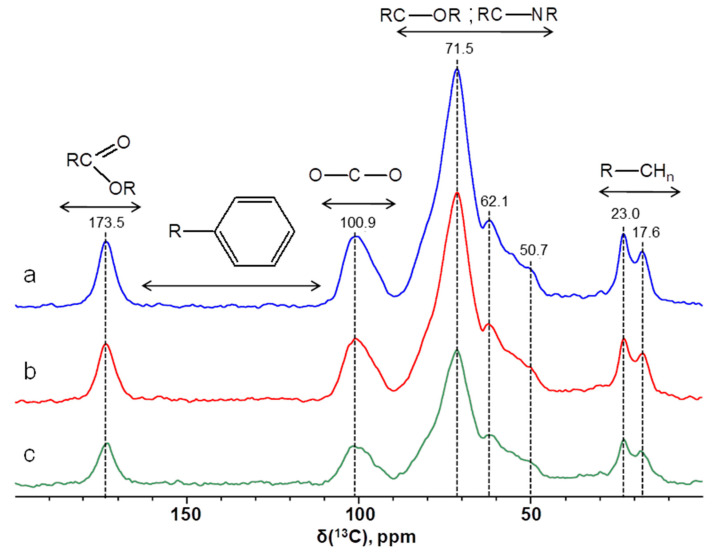
^13^C MAS NMR spectra of lyophilized HMWDOM extracted from a soil solution (**a**) Mw ≥ 14 kDa, (**b**) Mw ≥ 7 kDa, (**c**) Mw ≥ 3.5 kDa.

**Figure 2 plants-12-00649-f002:**
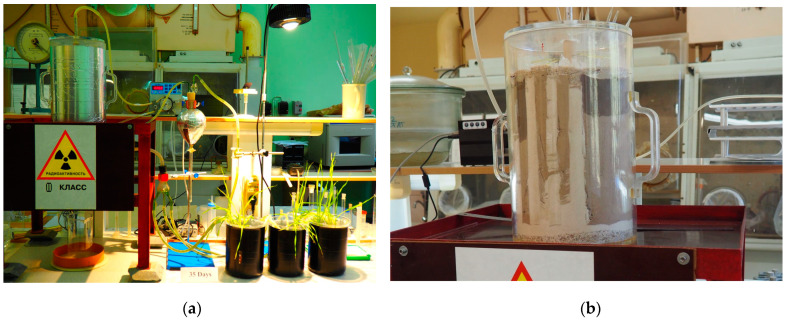
Experimental setup for studying Cd/^109^Cd migration parameters in the soil–lysimeter solution–plant system: (**a**) vegetation stand assembly; (**b**) lysimeter setup.

**Figure 3 plants-12-00649-f003:**
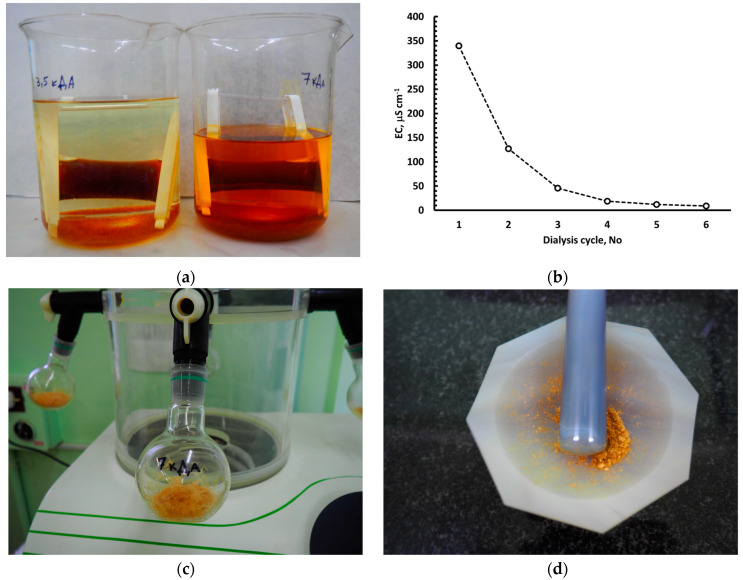
Successive steps of the preparation of a sample of dry HMWDOM-H from a soil solution: (**a**) Dialysis of the soil solution concentrate with MEMBRA-CEL dialysis bags with different pore sizes, (**b**) Control of the specific electrical conductivity (*EC*) of the sequential washings with easily soluble salts from the soil solution concentrate by dialysis; (**c**) Lyophilization of the HMWDOM dialysate of soil solution for subsequent analysis of its makeup by ^13^C MAS NMR; (**d**) Finished sample of lyophilized HMWDOM-H.

**Figure 4 plants-12-00649-f004:**
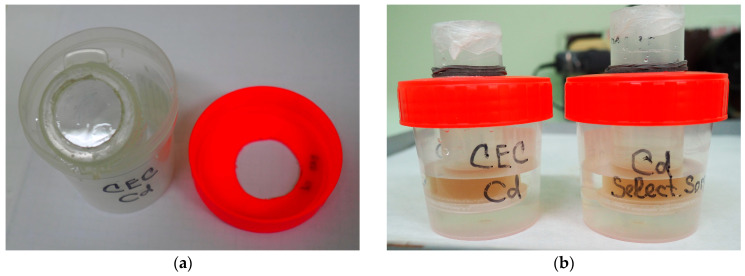
Determination of the cation exchange capacity (CEC) and selectivity coefficients *K_s_*^Cd/Ca^ of a HMWDOM sample (3.5 kDa): (**a**) The general view of a dialysis tube with a “window” made of a MEMBRA-CEL membrane (3.5 kDa) and an external container; (**b**) The completed assembly (inside the dialysis tubes there is a suspension of 30 mg of HMWDOM (3.5 kDa) + 10 cm^3^ of the equilibrating solution).

**Table 1 plants-12-00649-t001:** Values of major ^109^Cd (Cd) migration parameters in the soil–lysimeter solution–barley plant system per unit of dry weight (mean ± SD, n = 3).

Parameter	Value
A_m_(^109^Cd)_VP_, Bq kg^−1^	1665 ± 95
A_m_(^109^Cd)_root_, Bq kg^−1^	14,560 ± 1640
A_m_(^109^Cd)_RAC_, Bq kg^−1^	2950 ± 940
[Cd]_VP_, μg kg^−1^	430 ± 60
[Cd]_root_, μg kg^−1^	3100 ± 800
[Cd]_RAC_, μg kg^−1^	510 ± 100
*CF*(^109^Cd)_VP_, dm^3^ kg^−1^	1210 ± 70
*CF*(^109^Cd)_root_, dm^3^ kg^−1^	9500 ± 1400
*CF*(^109^Cd)_RAC_, dm^3^ kg^−1^	1890 ± 670
*CF*(Cd)_VP_, dm^3^ kg^−1^	1220 ± 120
*CF*(Cd)_root_, dm^3^ kg^−1^	10,800 ± 2100
*CF*(Cd)_RAC_, dm^3^ kg^−1^	1590 ± 320
*CR*(^109^Cd)_VP_, dm^3^ kg^−1^	3.35 ± 0.19
*CR*(^109^Cd)_root_, dm^3^ kg^−1^	29.2 ± 4.0
*CR*(^109^Cd)_RAC_, dm^3^ kg^−1^	5.9 ± 1.9
*CR*(Cd)_VP_, dm^3^ kg^−1^	2.02 ± 0.35
*CR*(Cd)_root_, dm^3^ kg^−1^	14.7 ± 2.8
*CR*(Cd)_RAC_, dm^3^ kg^−1^	2.35 ± 0.38
*A_sp_*(^109^Cd/Cd)_VP_, Bq μg^−1^	3980 ± 100
*A_sp_*(^109^Cd/Cd)_root_, Bq μg^−1^	4890 ± 1200
*A_sp_*(^109^Cd/Cd)_RAC_, Bq μg^−1^	5020 ± 840
S_active_, m^2^ g^−1^ (r.w.) *	0.18 ± 0.02
S_total_, m^2^ g^−1^ (r.w.)	0.67 ± 0.05
S_active_/ S_total_, %	27 ± 2
CEC_RAC_, meqv(+) kg^−1^(r.w.)	76 ± 28
SCD **, meqv(+) m^−2^(r.w.)	0.11 ± 0.04

* Raw weight.** Surface charge density.

**Table 2 plants-12-00649-t002:** ^13^C MAS NMR spectroscopy data for the most important structural elements of HMWDOM.

Range of Chemical Shifts, δ ^13^C ppm	Functional Group	Chemical Shifts of the Main Peaks in the Ranges, δ ^13^C ppm	Relative Contribution of Functional Group (Relative Spectra Intensity—I),%
3.5 kDa	7 kDa	14 kDa
160–190	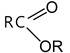 Carboxyl-C	173.5	8.2	7.6	7.5
113–160	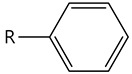 aromatic	-	0	0	0
93–113	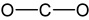 anomeric C	100.9	12.2	12.9	13.0
48–93	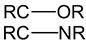 alkoxy: −O−C (alcohols), C-O-C (ethers) N-alkyl: amino acids	50.7; 62.1; 71.5	64.6	65.7	66.0
0–48	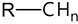 alkyl: −CH_3_; −CH_2_-; –CH<	17.6; 23	15.0	13.8	13.5

**Table 3 plants-12-00649-t003:** Parameters of specific activity Cd/^109^Cd in lysimeter waters, soil, VPs, roots and RAC.

Parameter	Equation
Specific activity (A_sp_) ^109^Cd/Cd in lysimeter solution, Bq µg^−1^	A_sp_(^109^Cd/Cd)_solution_ = A_V_(^109^Cd)_solution_/[Cd]_solution_
Specific activity (A_sp_) ^109^Cd/Cd in chemical fractions, Bq µg^−1^	A_sp_(^109^Cd/Cd)_Fr.#_ = A_m_(^109^Cd) _Fr.#_/[Cd]_Fr.#_
Specific activity (A_sp_) ^109^Cd/Cd in VPs, Bq µg^−1^	A_sp_(^109^Cd/Cd)_vp_ = A_m_(^109^Cd)_VP_ /[Cd]_VP_
Specific activity (A_sp_) ^109^Cd/Cd in roots, RAC, Bq µg^−1^	A_sp_(^109^Cd/Cd)_roots_ = A_m_(^109^Cd)_root, RAC_/[Cd]_root, RAC_

## Data Availability

Data sharing is not applicable to this article.

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
