# Peer review of "A Study on the Behavior of Cadmium in the Soil Solution–Plant System by the Lysimeter Method Using the 109Cd Radioactive Tracer"

_plants, 2023, doi:10.3390/plants12030649_

Round 1
Reviewer 1 Report
The manuscript presented interesting work on the distribution and transport of Cd in the soil-solution-plant system, reported some important experimental results, and gained some new insights. The reviewer felt that this manuscript was within the scope of the Journal and was ready for publication with minor revisions.
In line 13, the abbreviation Cd should appear after its full word, and in line 36, the same. Suggest using the abbreviation Cd for the word “cadmium” in other places in the text.
In the abstract, instead of just describing the experimental work, the major results and conclusions should be addressed.
In line 55, what is the “udic”?
In line 75, the abbreviation HM should appear after its full word.
In lines 139-141, it looks that “Cd stable/109Cd” should be “109Cd/Cd stable”.
In lines 201-202, “Δa” should be “Da”, similarly in Table 1.
In line 241, a subscript and a superscript were mislabeled.
In line 248, “и” should be “,”.
In the References part, there are either unnecessary space or typing errors at the beginning of items 11, 16, 17, 18, 19, 29, 33, 37, 43 and 45.
Author Response
We are grateful to the reviewer for his valuable comments. All the comments of the reviewer were taken into account during the revision of the manuscript: - mistakes and misprints that the reviewer drew attention to have been corrected; - in the Abstract, instead of an extended description of methodological techniques, we have given the major results and conclusions
Reviewer 2 Report
A good and interesting research, however, the manuscript writing should improved.
1. Cd or cadmium, should uniform;
2. Introduction, too much simple paragraphs, more attention should paid to the radioactive tracer, the current research progress for it;
3. Fig.2 c, longer error bar;
4. L249, so much “of”;
5. Discussion, too much simple paragraphs, more attention should paid to the transformation mechansim
6. Conclusion, the first graph was not necessary. Much were the results repetition.
Author Response
We are grateful to the reviewer for his valuable comments. All the comments of the reviewer were taken into account during the revision of the manuscript: - we have paid more attention in the Introduction chapter to the use of radiotracers to assess the mobility of heavy metals in soils and the current research progress for it; - mistakes and misprints that the reviewer drew attention to have been corrected; - in the Discussion chapter we have paid more attention to the mechansim of Cd forms transformation in a soil; - figures 1 and 2 were deleted because they duplicated the material given in the text of the manuscript.
Reviewer 3 Report
I have completed the review of the manuscript under reference. Authors have investigated lability of Cd using lysimetric methods. The study is important in view of understanding the soil factors affecting Cd availability in the soil. I have some suggestions as under:
1. Authors need to provide an overview of the Cd content in various soil types. Further, what are the different factors that have already been involved/implicated in metal availability? How is the present study different and what new information is presented?
2. Introduction section be revised, and information be collated in fewer paragraphs.
3. Sub-heads be revised and shortened.
4. Conclusion be shortened and should describe main findings and should not summarize the results!
Author Response
We are grateful to the reviewer for his valuable comments. All the comments of the reviewer were taken into account during the revision of the manuscript:
- we have provided an overview of the Cd content in various soil types;
- the description of the different factors that have been involved in metal soil mobility and bioavailability was given;
- introduction section has been seriously revised;
- sub-heads have been revised and shortened;
- conclusion be shortened and the attention in this chapter was focused on describing of main findings.

Reviewer 4 Report
The MS has an orientation for Chemistry readers. Not much is there for plant biologists.
Nothing is explained about plant,s nature, how it has been raised, nutrition, growing conditions etc. Moreover, nothing has been said about its growth, parameters wrf to Cd. I will suggest that the authors should submit it to some journal having a Chemistry background. Not suitable for "PLANT"
English also needs greater improvement before submitting to any other journal.
Author Response
We are grateful to the reviewer for a value comment about the insufficient coverage in the manuscript of the moments related to the physiological aspects of Cd migration. We corrected this omission and supplemented the material of the article with the definition of a number of important biological parameters characterizing the process of Cd absorption and uptake by barley roots from an equilibrium lysimeter solution. Researchers usually do not pay due attention to these key parameters. However, they are extremely important in modeling heavy metals migration (including Cd) in the soil–soil solution system and root uptake from the latter. These are the following additional parameters (to those already given in the manuscript): the active and total surface area of barley roots, the ion exchange capacity of the root absorbing complex (RAC), the specific mass and surface densities of ionogenic groups in RAC, the concentration factors of stable and radioactive cadmium isotopes in the apoplast.

Round 2
Reviewer 4 Report
The authors have incorporated all the suggestions and improved the MS